# Non-Target Metabolomics Reveals Changes in Metabolite Profiles in Distant Hybrid Incompatibility Between *Paeonia* sect. *Moutan *and* P. lactiflora*

**DOI:** 10.3390/plants14091381

**Published:** 2025-05-03

**Authors:** Wenqing Jia, Yingyue Yu, Zhaorong Mi, Yan Zhang, Guodong Zhao, Yingzi Guo, Zheng Wang, Erqiang Wang, Songlin He

**Affiliations:** 1School of Horticulture and Landscape Architecture, Henan Institute of Science and Technology, Xinxiang 453003, China; yuyingyue0923@163.com (Y.Y.); mizr@hist.edu.cn (Z.M.); mint_19@163.com (Y.Z.); 2Luoyang National Peony Gene Bank, Luoyang 471011, China; zhaoguodong2025@163.com; 3College of Landscape Architecture and Art, Henan Agricultural University, Zhengzhou 450002, China; guoyz2017@126.com (Y.G.); wzhengt@163.com (Z.W.); 4Luoyang Academy of Agricultural and Forestry Sciences, Luoyang 471099, China; 13629805339@163.com

**Keywords:** stigma metabolite, nucleotide metabolism, purine metabolism, nucleotide sugar metabolism, metabolite analysis

## Abstract

Peonies are globally renowned ornamental plants, and distant hybridization is a key method for breeding new varieties, though it often faces cross-incompatibility challenges. The metabolic mechanisms underlying the crossing barrier between tree peony (*Paeonia* sect*. Moutan*) and herbaceous peony (*P. lactiflora*) remain unclear. To identify key metabolites involved in cross-incompatibility, we performed a cross between *P. ostii* ‘Fengdanbai’ (female parent) and *P. lactiflora* ‘Red Sara’ (male parent) and analyzed metabolites in the stigma 12 h after pollination using UPLC-MS. We identified 1242 differential metabolites, with 433 up-regulated and 809 down-regulated, including sugars, nucleotides, amino acids, lipids, organic acids, benzenoids, flavonoids, and alkaloids. Most differential metabolites were down-regulated in hybrid stigmas, potentially affecting pollen germination and pollen tube growth. Cross-pollinated stigma exhibited lower levels of high-energy nutrients (such as amino acids, nucleotides, and tricarboxylic acid cycle metabolites) compared to self-pollinated stigma, which suggests that energy deficiency is a contributing factor to the crossing barrier. Additionally, cross-pollination significantly impacted KEGG pathways such as nucleotide metabolism, purine metabolism, and vitamin B6 metabolism, with most metabolites in these pathways being down-regulated. These findings provide new insights into the metabolic basis of cross-incompatibility between tree and herbaceous peonies, offering a foundation for overcoming hybridization barriers in peony breeding.

## 1. Introduction

Tree peony (*Paeonia* sect*. Moutan*) is one of the top ten traditional and famous flowers in China, and it is important for its horticultural, ornamental, medicinal, and cultural value [1,2]. As an important means of plant breeding, crosses between different species or genera can create new varieties with excellent characteristics [3]. Distant hybrids between *P.* sect*. Moutan* and *P. lactiflora* often use *P. lactiflora* as the female parent and *P. sect. Moutan* as the male parent, with Itoh peonies being a typical example of this hybridization method, successfully combining the disease resistance of herbaceous peony with the ornamental value of tree peony [4]. However, when tree peony is used as the female parent and herbaceous peony as the male parent for distant hybridization, severe hybridization barriers are often encountered [5]. This is due to the distant hybridization incompatibility between tree peony and herbaceous peony, which is manifested as low pollen germination rates, abnormal pollen tube growth, fertilization failure, etc. [6,7]. Hybrid incompatibility has also been extensively reported in other plant species. For example, Gong et al. [8] studied unilateral incompatibility between *Camellia yuhsienensis* and *Camellia oleifera* and found that in incompatible combinations, the pollen tubes grew slowly and exhibited swelling when reaching the base of the style. Similarly, Hao et al. [9] studied the effects of different pollination methods on the hybridization compatibility of the azalea and the *Rhododendron decorum* and also observed that there were twists and tangles during pollen tube growth. Erandi et al. [10] studied the hybridization of triploid watermelon, discovering that a significant number of pollen grains failed to germinate and that the pollen tubes of most germinated pollen grains were twisted. Kuligowska et al. [11] studied interspecific hybridization in the *Kalanchoë* genus and found anomalous pollen tubes with branching phenomena and disordered growth direction. Ferriol et al. [12] studied interspecific hybridization barriers in *Cucumis* and found that incompatible pollen tubes stopped growing upon reaching the stigma. These studies suggest that distant hybridization incompatibility may involve complex interaction mechanisms between pollen and stigma, affecting the growth of incompatible pollen tubes.

The stigma, as the apical part of the pistil, plays a crucial role in pollen germination and pollen tube growth. In the process of distant hybrid pollination, the interaction between stigma and pollen is one of the major elements affecting hybridization success [13,14]. Metabolites, as direct products of biochemical reactions in organisms, are important indicators of the physiological state and metabolic activities of cells [15]. Metabolomics technology has developed rapidly in recent years and has become an important tool for studying complex biological problems such as plant adversity response, biotic invasion defense, and genetic variation [16,17,18]. Metabolite analysis can comprehensively and systematically resolve the metabolic network of plants under different conditions, revealing the dynamics of change in key pathways and providing new perspectives for understanding metabolic regulatory mechanisms during plant reproduction [19]. Therefore, studying stigma metabolites may help to elucidate the physiological mechanism of the distant crossing barrier in tree peony.

Previous studies have identified metabolite changes associated with both self-incompatibility and cross-incompatibility in some plants. For example, the levels of lipids, organic acids, amino acids, and their derivatives are significantly altered in self-incompatible flowers of *Acacia mangium* [20]. The sweet potato stigma metabolome revealed that flavonoids are the key metabolites regulating intraspecific cross-incompatibility in sweet potato [21]. Multi-omics analysis of oil tea pistils showed that high levels of galloylated catechins contribute to the inhibition of self-pollen tubes in oil tea [22]. The metabolomic analysis of hybrid ovules between *Hydrangea macrophylla* and *H. arborescens* has revealed that salicylic acid plays a pivotal role in embryo abortion [23].

In our previous study, we found apparent pollen–pistil incompatibility between the inter-sectional hybrid between *P. ostii* ‘Fengdanbai’ and *P. lactiflora* ‘Red Sara’ [24]. In this study, we conducted metabolomic analysis on peony stigma 12 h after pollination, with the aim of (1) comparing the effects of cross-pollination and self-pollination on the growth of pollen tubes, including pollen tube length, growth rate, and morphological changes; (2) combining physiological indicators and metabolite changes and further exploring the potential links between these metabolic changes and physiological indicators; and (3) comparing the metabolite changes in hybrid and self-cross stigma after pollination and exploring the biomarker metabolites associated with hybrid incompatibility.

## 2. Results and Discussion

### 2.1. Effect of Cross-Pollination on Pollen Tube Growth

As shown in Figure 1, 12 h after pollination, a substantial amount of pollen germination was observed in the CK, and the pollen tube growth direction was consistent, with the pollen tubes extending along the style and reaching the mid-to-lower part of the style in CK (Figure 1a). At this time, a small amount of pollen germinated in T12, and these pollen tubes were twisted and distorted on the surface of the stigma, with very few pollen tubes growing into the interior of the stigma. Meanwhile, the deposition of callose was observed in T12 (Figure 1b).

At 24 h after pollination, most of the pollen tubes grew toward the ovary in the CK, and the pollen tubes reached the junction of the style and the ovary (Figure 1c). In the meantime, some pollen tubes grew in the opposite direction to the style in the T24; a very small proportion of pollen tubes can continue to elongate normally to 1/2 to 2/3 of the style. It can be observed that some pollen tubes are short and disorderly in direction in T24 (Figure 1d). These phenomena indicate that the growth of hybrid pollen tubes faces significant challenges. Based on these observations, we selected stigma from both hybrid and self-pollination at 12 h post-pollination for metabolomics studies, aiming to investigate the changes in the metabolic substances underlying the differences in pollen tube growth.

### 2.2. Changes in ZR, IAA, ABA, BR, MeJA, and MT Contents

As can be seen in Figure 2, the contents of zeatin riboside (ZR) and indole-3-acetic acid (IAA) in hybrid stigmas are significantly lower than those of the CK. Meanwhile, the contents of abscisic acid (ABA), brassinosteroid (BR), methyl jasmonate (MeJA), and melatonin (MT) are significantly higher than those of the CK at 12 h after pollination.

ZR, a cytokinin (CTK), plays a crucial function in cellular division and proliferation [25]. Breygina et al. [26] found that the level of cytokinin was greatly increased after the polar growth of the pollen tubes of both spruce and tobacco plants. Zhao et al. [27] found that the CTK levels in *Prunus armeniaca* ‘Li Guang’ aborted flower exceeded those of normal blooms. In this study, the decrease in ZR content in the hybrid stigmas may have inhibited cell division and elongation of the pollen tube, resulting in slow growth and a decrease in the number of germinations.

IAA, being the predominant growth hormone in plants, is crucial for their development [28]. The decrease in IAA content may also be detrimental to the normal growth of pollen tubes. The reduction in both ZR and IAA may have caused the abnormalities in pollen tube growth and germination in hybridization incompatibility.

ABA is frequently linked to plant stress responses and growth suppression [29]. In hybrid stigmas, the elevated ABA concentration may indicate the stigma’s rejection response to hybrid pollen. BR participate in numerous plant growth and developmental processes [30], and appropriate concentrations of BR enhance pollen germination and growth in *Arabidopsis thaliana* [31]. However, the significant increase in BR content may have exceeded the appropriate range in this study, resulting in the inhibition of pollen tube growth. This may be related to the physiological disorders caused by hybridization incompatibility.

The increase in MeJA may reflect the defensive response of stigma to hybrid pollen. Zhao et al. [32] found that an appropriate concentration of MeJA can promote the germination of camellia pollen and the growth of pollen tubes, but high concentrations inhibit growth. In this study, the significant increase in MeJA content in the stigma may have adversely affected the hybrid pollen tube growth. A previous study on *Pinus nigra* found that MeJA can alter the direction of pollen tube growth by influencing actin structure and callus distribution [33]. This may explain the disorder of pollen tube orientation in hybrids. Melatonin shares a similar structure and metabolic pathway with IAA [34]. Qi et al. [35] found that melatonin can reduce the inactivation of tomato pollen and the inhibition of pollen germination caused by high temperatures. Hu’s [36] research on cotton pollen fertility found that exogenous melatonin can improve the viability of cotton pollen under drought stress. In this study, the significant increase in melatonin content in T12 may be a response of the stigma to the rejection of hybrid pollen.

### 2.3. Metabolome Analysis

LC-MS/MS was employed to analyze the metabolite composition of peony stigma 12 h post-pollination in the T12 and CK groups. As shown in Figure 3, the six samples were divided into two different groups using PCA (Principal Component Analysis). The differences between T12 and CK were mainly explained by PC1 and PC2 in the model, which accounted for 65.87% and 9.25% of the variables, respectively. This experiment performed univariate and multivariate statistical methods to identify difference metabolites (VIP > 1, |log2FC| ≥ 1, and *p* < 0.05) between T12 and CK. At 12 h after pollination, T12 and CK had 1242 divergent metabolites, of which 433 were up-regulated and 809 down-regulated.

The raw relative amounts of the differential metabolites discovered by employing the screening criteria were standardized by rows using Unit Variance Scaling (UV), and heat maps were generated using the Complex Heatmap package in R software. A total of 1242 differential metabolites were categorized into nine classes, including sugars, nucleotides, amino acids, lipids, organic acids, benzenoids, flavonoids, alkaloids, and others.

Significant metabolic differences in incompatible stigma were revealed by orthogonal partial least squares discriminant analysis (OPLS-DA) (Figure 4). The top 20 metabolites with VIP values that were up-regulated included the phenylpropane metabolites p-coumaroylagmatine (VIP = 1.24) and L-phenylalanine (VIP = 1.24) and the oxidative stress-related metabolites gluconolactone (VIP = 1.24) and pyridoxamine (VIP = 1.24), and those that were down-regulated include adenine (VIP = 1.24), guanine (VIP = 1.24), and cyclic AMP (VIP = 1.23) of the nucleotide metabolism pathway and zeatin biosynthesis pathway of Indole-3-acetamide, etc.

#### 2.3.1. Changes in Metabolic Pathways

Metabolic pathway analysis (Figure 5) showed that cross-pollination significantly altered the metabolic pathways of nucleotide metabolism, purine metabolism, zeatin biosynthesis, nucleotide sugar biosynthesis, amino acid sugar and nucleotide sugar metabolism, and cofactor biosynthesis in the stigma.

Nucleotide sugars are one of the precursor substances for plant cell wall synthesis, and in the process of pollen tube cell wall formation, nucleotide sugars, such as UDP-glucose, are converted into polysaccharide substances that constitute the cell wall, including cellulose and pectin, through a series of enzyme-catalyzed reactions [37]. Figure 6 shows that the biosynthesis pathway of nucleotide sugars is inhibited after cross-pollination, which may lead to a decrease in the synthesis of cell wall polysaccharides, thereby affecting the growth rate, morphology, and direction of pollen tubes. Meanwhile, the down-regulation of most metabolites in the zeatin biosynthesis pathway in the hybrid stigma resulted in the reduced synthesis of cytokinins in the stigma, which may affect the growth rate of pollen tubes. Figure 6 also shows that cross-pollination significantly affected the metabolic pathways in the stigma, with the most pronounced changes observed in purine and nucleotide metabolism. Among the 15 differential purine metabolites, 11 were simultaneously involved in nucleotide pathways (*p* < 0.05), which suggests coordinated regulation between these two metabolic pathways. Five purine nucleotides exhibited significant up-regulation: adenosine-5′-diphosphate (ADP, 3.75-fold increase), inosinic acid (IMP, 3.03-fold increase), guanosine-5′-diphosphate (GDP, 2.90-fold increase), xanthosine-5′-monophosphate (XMP, 2.68-fold increase), and 2′-deoxyguanosine-5′-diphosphate (dGDP, 2.43-fold increase). By contrast, the purine bases adenine and guanine decreased by 3.85-fold and 3.70-fold, respectively. The accumulation of purine nucleotides alongside the decrease in purine bases may indicate that after cross-pollination, the stigma enhances nucleotide biosynthesis to support the energy and nucleic acid synthesis required for pollen tube penetration.

As shown in Figure 7 and Appendix A, of the 1242 differential metabolites detected on the stigma, 108 were annotated to the KEGG pathway. Among them, nucleotide metabolites were the most numerous (27), followed by alkaloids (13), organic acids (11), amino acids (9), sugars (8), and lipids (7). Changes in these metabolites may have important effects on the physiological state of the stigma.

#### 2.3.2. Changes in Sugars

At 12 h after pollination, 32 differential sugars were identified in the stigma, of which 13 were up-regulated, including gluconolactone (log2FC = 1.93), digalacturonate (log2FC = 1.69), xylobiose (log2FC = 1.37), d-mannose (log2FC = 2.18), galactinol (log2FC = 3.55), and alpha-d-galactosamine 1-phosphate (log2FC = 1.72). The 19 down-regulated differential sugars included 2-acetamido-2-deoxy-d-mannopyranose (log2FC = −5.06), mannose-6-phosphate (log2FC = −1.43), and others (Appendix A).

Sugars are important energy substances as well as signaling molecules that play an irreplaceable role in plant reproduction [38]. Pollen grain germination utilizes stored carbohydrates (mainly starch), while the rapid growth of pollen tubes is a high ATP-consuming process that requires the intake of external carbohydrates (mainly sucrose) [39]. In this study, the down-regulation of metabolites in the starch and sucrose pathways may have affected the supply of external sucrose required for pollen growth, resulting in reduced pollen hydration and germination. In *Arabidopsis*, excess D-mannose and galactose inhibit pollen germination by interfering with sugar metabolism [40,41]. In the present study, we found that mannose content was significantly up-regulated in peony stigmas, which may produce a similar inhibitory effect and, thus, adversely affect pollen germination.

In addition, carbohydrates constitute a significant component of plant cell walls. Among them, digalacturonate is a product of pectin degradation [42]. Dewangan et al. [43] found that xylobiose treatment of *Arabidopsis thaliana* enhanced the ROS response and induced callose deposition, altering cell wall composition and hormone levels. These metabolites may affect the penetration of hybrid pollen tubes into the stigma by affecting the structure of the stigma cell wall.

#### 2.3.3. Changes in Amino Acids

At 12 h after pollination, there were 418 differential amino acids identified in the stigma, of which 117 were up-regulated and 301 were down-regulated. The up-regulated amino acids included L-phenylalanine (log2FC = 1.88), nicotinuric acid (log2FC = 2.12), L-alpha-aminobutyric acid (GABA, log2FC = 1.30), L-proline (log2FC = 1.07), and AICA ribonucleotide (log2FC = 1.29). Down-regulated amino acids included L-glutamine (log2FC = −1.09), L-aspartic acid (log2FC = −1.64), 4-(phosphonooxy)-L-threonine (log2FC = −1.40), and 5-phosphonooxy-L-lysine (log2FC = −1.64) (Appendix A).

It has been reported that amino acids produced by protein hydrolysis are substrates for pollen tube energy supply. Changes in the amino acid pool may be the result of physiological changes in the pollen tube and may also further regulate pollen tube growth by affecting energy supply [44]. In hybrid stigmas, the down-regulation of most amino acids may lead to an insufficient energy supply, which may be one of the possible reasons affecting the growth of hybrid pollen tubes. Wang et al. [45] highlighted that *Arabidopsis* pollen tube growth was inhibited by high concentrations of GABA. In addition, studies in tomato pollen have shown that during plant reproduction under non-stress conditions, GABA plays a crucial role in facilitating proper pollen tube formation by regulating calcium-permeable membrane channels that generate the calcium gradient that guides pollen tube elongation [46]. Therefore, the up-regulation of GABA may affect the polar growth of hybrid pollen tubes by disrupting calcium gradients.

#### 2.3.4. Changes in Nucleotides

At 12 h after pollination, 47 differential amino acids were identified in the stigma, with 19 exhibiting up-regulation and 28 demonstrating down-regulation. The up-regulated ones were ADP (log2FC = 1.91), XMP (log2FC = 1.42), IMP (log2FC = 1.60), GDP (log2FC = 1.54), thymidine (log2FC = 2.50), diguanosine tetraphosphate (log2FC = 4.17), 7-methylxanthine (log2FC = 1.60), 2′-Deoxyguanosine-5′-diphosphate (log2FC = 1.28), and AICA ribonucleotide (log2FC = 1.29). Down-regulated ones included adenosine monophosphate (AMP, log2FC = −1.69), uridine-5′-diphosphate (UDP, log2FC = −2.00), uridine-5′-monophosphate (UMP, log2FC = −2.06), UDP-glucose (UDP-Glc, log2FC = −1.32), UDP-xylose (UDP-Xyl, log2FC = −2.32), UDP-L-arabinofuranose (log2FC = −1.47), nicotinic acid adenine dinucleotide (NAD+, log2FC = −1.03), cyclic AMP (cAMP, log2FC = −1.03), guanine (log2FC = −1.89), adenine (log2FC = −1.94), cytidine (log2FC = −1.64), adenosine (log2FC = −1.60), and guanosine (log2FC = −1.84) (Appendix A).

The down-regulation of pyrimidine nucleotides, such as UMP, UDP, and UDP-Glc, and monophosphate purines, such as AMP, in hybrid stigma may reflect a disturbance in energy metabolism, which may be a manifestation of insufficient energy supply in hybrid stigma. The limitation of pyrimidine nucleotides is often accompanied by changes in purine levels and metabolic processes, including nucleotide metabolism, intracellular transport, carbohydrates, and energy metabolism [47].

UDP-Glc is a substrate for glycosyltransferases and is involved in the synthesis of cell wall polysaccharides in pollen tubes [48]; UDP-Xyl is an important component of xyloglucan and xylan and plays a key role in cell wall extension [49]. Therefore, the down-regulation of UDP-Glc and UDP-Xyl may hinder the growth of hybrid pollen tubes by affecting the synthesis and extension of the cell wall.

Nucleotide metabolism is not only involved in substance metabolism but also closely related to intracellular signaling. For example, nicotinic acid adenine dinucleotide phosphate (NAADP) is a widely present second messenger, and NAD+ is its breakdown product [50]. Therefore, down-regulation of NAD+ may imply that intracellular signal transmission in T12 is disturbed, thus affecting signal recognition and transmission. cAMP is a signaling molecule involved in the reorientation of pollen tubes, and it changes the direction of pollen tube growth by its transient elevation of the apical region [51]. cAMP can activate Ca^2^⁺ channels in pear pollen [52]. In this study, the down-regulation of cAMP in hybrid stigma may lead to a disordered calcium gradient, affecting the polar growth of pollen tubes and causing chaotic growth direction.

#### 2.3.5. Changes in Lipids

At 12 h after pollination, there were 21 differential lipids identified in the stigma, of which four were up-regulated and 17 were down-regulated. Among the down-regulated were 9,10-dihydroxystearic acid (log2FC = −1.25), sphingosine-1-phosphate (log2FC = −1.79), and floionolic acid (log2FC = −1.12). The up-regulated lipids included lysophosphatidylcholine 16:0 (log2FC = 1.46) and 16-hydroxyhexadecanoic acid (log2FC = 1.30) (Appendix A).

Lipids form a hydrophobic barrier on the surface of pollen grains, regulating water uptake and retention [49,53]. The reduction of most lipids in T12 may have weakened this barrier function, resulting in inadequate water uptake by the pollen and the appearance of shriveled pollen. In addition, 16-hydroxyhexadecanoic acid is enriched in cutin, cork, and wax biosynthesis; 16-hydroxyhexadecanoic acid is one of the major C16 and C18 fatty acid-derived cutin monomers [54]. Up-regulation of 16-hydroxystearic acid in the stigma can lead to thickening or structural changes in the cuticle, which makes it difficult for the cuticle to break down. This may prevent the pollen tube from penetrating the stigma and affect the normal growth of the pollen tube.

#### 2.3.6. Changes in Flavonoids

At 12 h after pollination, 61 differential flavonoids were present in the stigma, of which 21 were up-regulated and 40 were down-regulated. The down-regulated flavonoids included taxifolin (log2FC = −1.64) and neohesperidin (log2FC = −1.40). The up-regulated flavonoids included naringenin (log2FC = 2.32) (Appendix A).

Quercetin has a role in activating pollen tube germination in *Arabidopsis,* and taxifolin is structurally similar to quercetin [38,55]. Down-regulation of taxifolin may weaken the germination ability of pollen tubes and affect the initial growth of pollen tubes. The naringenin chalcone isomerized to naringenin via chalcone isomerase, and the chalcone isomerase-deficient mutant exhibits a phenotype of inhibited pollen tube growth and compromised pollen tube integrity [56]. Naringenin synthesizes various flavonoids with the help of other enzymes in the flavonoid biosynthesis pathway. Flavonols are one of the final products of the flavonoid biosynthesis pathway and are associated with the penetration of rice pollen tubes into the stigma [57]. Cross-pollination may have disrupted the homeostasis of flavonoids in the stigma, reducing pollen tube growth.

#### 2.3.7. Changes in Alkaloids

At 12 h after pollination, 63 differential alkaloids were present in the stigma, of which 25 were up-regulated and 38 down-regulated. Among the down-regulated ones were indole-3-acetamide (log2FC = −1.84), Carnitine C16:0 (log2FC = −2.64), indole (log2FC = −1.15), dihydromacarpine (log2FC = −2.25), camptothecin (log2FC = −2.40), dihydrosanguinarine (log2FC = −2.40), and 19-hydroxytabersonine (log2FC = −1.15). Up-regulated ones included pyridoxamine (log2FC = 1.01), p-coumaroylagmatine (log2FC = 1.55), imidazoleacetic acid (log2FC = 2.40), agmatine (log2FC = 1.77), harmaline (log2FC = 1.43), D-(+)-Neopterin (log2FC = 2.50), and diethanolamine (log2FC = 1.13) (Appendix A).

Pérez-Alonso et al. [58] found that in Arabidopsis, indole-3-acetamide (IAM) is converted to the major plant growth hormone indole-3-acetic acid (IAA) by amidase 1 (AMI1). Plants lacking AMI1 exhibit growth inhibition and enhanced ABA accumulation. In this study, the down-regulation of IAM in the hybrid stigma may have disrupted the homeostasis of IAA and ABA, thereby adversely affecting pollen tube growth. In addition, Lu et al. [59] showed that salt stress promotes the accumulation of pyridoxamine (VB6) in maize roots to scavenge excess ROS or induce the accumulation of ABA. The up-regulation of VB6 in the hybrid stigma in this study may indicate that the stigma was subjected to oxidative stress, which, in turn, affected the growth environment of the pollen tube. Agmatine is a good precursor of spermine [60]. A study in tea (*Camellia sinensis*) found that low concentrations of putrescine can promote the germination of tea pollen and pollen tube elongation, but when the concentration exceeds 400 μM, it can inhibit pollen tube growth by affecting the distribution of sucrose synthase and changing the arrangement of actin filaments in the pollen tube [61]. The up-regulation of agmatine in the hybridized stigma in this study may have interfered with the growth direction of the pollen tube through a similar mechanism, ultimately affecting the fertilization process.

#### 2.3.8. Changes in Organic Acids

At 12 h after pollination, 150 different organic acids were identified in the stigma, of which 62 showed up-regulation and 88 down-regulation. Among the down-regulated organic acids were chorismic acid (log2FC = −2.84), (2S)-2-(3-carboxypropanamido)-6-oxoheptanedioic acid (log2FC = −2.47), geranyl diphosphate (log2FC = −1.47), (2S)-2-isopropylmalate (log2FC = −1.22), maleic acid (log2FC = −1.51), hydrogenobyrinic acid (log2FC = −3.18), and trypanothione (log2FC = -1.06). Up-regulated organic acids included 3-(methylthio) propionic acid (log2FC = 1.82), 2-amino-4-oxovaleric acid (log2FC = 1.81), isonocardicin A (log2FC = 1.49), and isopentenyl pyrophosphate (log2FC = 1.24) (Appendix A).

Of particular interest is (2S)-2-isopropylmalate (2-IPMA), a key intermediate in leucine biosynthesis [62]. Leucine metabolism plays a critical role in pollen development, as evidenced by studies showing that *Arabidopsis* mutants deficient in leucine-rich repeat elongation proteins have impaired pollen germination and pollen tube growth [63]. In this study, the down-regulation of 2-IPMA in hybrid stigma suggests a possible disruption in leucine biosynthesis, which may have contributed to the observed inhibition of hybrid pollen tube elongation and growth.

## 3. Materials and Methods

### 3.1. Materials

The experiment was carried out in mid-April 2023 in the experimental field of Henan Institute of Science and Technology (Xinxiang, China). The female parent was *P. ostii* ‘Fengdanbai’. The self-pollinated pollen was collected from the green space in front of Building Zero at Henan Institute of Science and Technology, where the microclimate conditions induced flowering approximately one week earlier compared to the experimental field. The male parent wa*s P. lactiflora* ‘Red Sara’. The pollen was obtained from forced-flowering plants at Luoyang Flower Market, collected around the Spring Festival in 2023 and preserved at −80 °C in an ultra-low temperature freezer after desiccation treatment. At the same time, pollen from ‘Fengdanbai’ was collected for self-pollination as the control group (CK), and cross-pollination between ‘Fengdanbai’ and ‘Red Sara’ was used as the experimental group (T). Stigmas were collected 12 h after pollination (T12), and each treatment was repeated three times. Each sample weighed 0.5 g and was immediately placed in liquid nitrogen and then stored in an ultra-low temperature refrigerator at −80 °C. The samples were sent to Metware Metabolism Company (Wuhan, China) for non-targeted metabolome assay.

### 3.2. Methods

#### 3.2.1. Pollen Tube Fluorescence Observation

Pistils were preserved in Carnot solution (anhydrous ethanol/glacial acetic acid [3:1]) for 24 h, then transferred to 70% ethanol and stored at 4 °C for backup. Prior to pressing, the fixed pistils were softened in 8 mol/L NaOH at 60 °C for 4 h, followed by staining with 0.5% water-soluble aniline blue in 1 mol/L KH_2_PO_4_ solution, shielded from light for 6 h. The pistil was unfolded lengthwise and then pressed under a fluorescence microscope to observe pollen tube growth and fertilization.

#### 3.2.2. Determination of Endogenous Hormone Content

The concentrations of zeatin riboside (ZR), indole-3-acetic acid (IAA), abscisic acid (ABA), brassinosteroid (BR), methyl jasmonate (MeJA), and melatonin (MT) were quantified utilizing the corresponding enzyme-linked immunosorbent assay (ELISA) kits. Results are expressed as mean ± SE of three replicates. Statistical analysis was performed using one-way ANOVA in SPSS 23.0, with *p* < 0.05 considered significant.

#### 3.2.3. Metabolite Identification

Six stigma samples (three each from T12 and CK groups, 0.2 g per sample) were vacuum freeze-dried in a freeze-dryer (Scientz-100F, Ningbo Scientz Biotechnology, Ningbo, China) and ground into powder using a grinding mill (MM 400, Retsch GmbH, Haan, Germany) at a frequency of 30 Hz for 1.5 min. Then, 50 mg of the powdered sample (weighed using an electronic balance MS105DM) was mixed with 1200 μL of a pre-cooled (−20 °C) 70% methanol aqueous solution containing internal standard extraction solvent. The mixture was intermittently vortexed (vortexed for 30 s every 30 min, repeated 6 times) and then centrifuged at 12,000 rpm for 3 min. The supernatant was filtered through a 0.22 μm membrane and used for UPLC-MS/MS analysis.

Metabolite analysis was performed using an ultra-performance liquid chromatography system (Shimadzu LC-30A, Shimadzu Corporation, Kyoto, Japan) with a Waters ACQUITY UPLC HSS T3 column (1.8 µm, 2.1 mm × 100 mm, Waters Corporation, Milford, MA, USA). The main liquid-phase conditions were as follows: column— mobile phase A: ultrapure water (0.1% formic acid); mobile phase B: acetonitrile (0.1% formic acid); instrumental column temperature: 40 °C; flow rate: 0.4 mL/min; and injection volume: 4 μL. The main MS conditions were as follows: MS analysis—metabolite extraction, detection, data pre-processing, data control, and statistical analysis. Mass spectrometry analysis was performed using a SCIEX TripleTOF 6600+ mass spectrometer (SCIEX, Framingham, MA, USA) with the following parameters: ion source temperature (ESI) 550 °C, ionization voltage (ESI+) 5500 V, and spray gas 50 psi.

The mass spectrometry downcomer raw data were converted to mzXML format by ProteoWizard, and the XCMS program was used for peak extraction, alignment, and retention time correction. The peaks with >50% missing rate in each group of samples were filtered, and the blanks were filled with K-nearest neighbor (KNN), and the peak areas were corrected by support vector regression (SVR). The corrected and filtered peaks were used for metabolite identification by searching the self-constructed database, integrating public libraries and predictive libraries, and mitochondrial DNA (metDNA) method in the laboratory of Metware Metabolism Company (Wuhan, China).

In our study, fold change (FC) was calculated by dividing the mean value of metabolite content in the T12 group by the mean value of metabolite content in the CK group. Metabolites with a |log2FC| ≥ 1, *p* < 0.05, and a variable importance in projection (VIP) value > 1 were identified as differential metabolites.

The differential metabolites detected were subjected to data pre-processing and data quality control, and the obtained high-quality data were statistically analyzed to generate clustered heat maps, KEGG pathway enrichment classifications, and scatter plots.

## 4. Conclusions

In this study, *P. ostii* ‘Fengdanbai’ was used as the female parent and *P. lactiflora* ‘Red Sara’ as the male parent for hybridization. Metabolomics analysis of stigma at 12 h after pollination revealed the effect of hybridization on pollen tube growth and its associated metabolic network. Fluorescence observation of pollen tubes showed that hybrid pollination significantly inhibits the growth of pollen tubes, which manifests as reduced pollen tube germination rates, disorganized growth directions, and restricted elongation. Metabolomic analysis showed that hybrid pollination significantly altered key metabolic pathways in the stigma, including nucleotide metabolism, purine metabolism, zeatin biosynthesis, and nucleoside sugar biosynthesis. After distant hybridization pollination, the significant down-regulation of most differential sugars, amino acids, nucleotides, lipids, flavonoids, and alkaloids disrupts stigma energy supply, pollen hydration, and polar growth, leading to hybridization barriers. This study delves into the effects of distant hybridization pollination between tree peony and herbaceous peony on the metabolites of the stigma, laying a foundation for elucidating the regulatory network of metabolites during the hybridization process.

## Figures and Tables

**Figure 1 plants-14-01381-f001:**
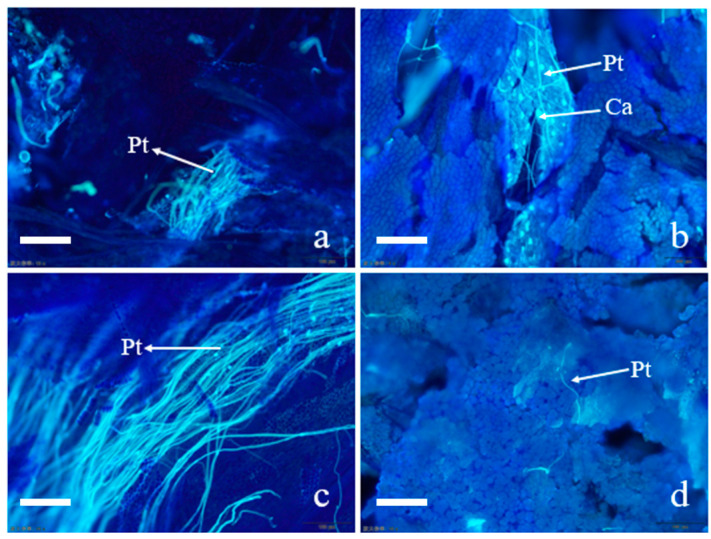
Pollen grain germination and pollen tube growth after self-pollination and cross-pollination of *Paeonia ostii* ‘Fengdanbai’. (**a**) Pollen tubes in self-crossed stigma at 12 h after pollination; (**b**) pollen tubes in hybrid stigma at 12 h after pollination; (**c**) pollen tube in self-crossed stigma at 24 h after pollination; (**d**) pollen tubes in hybrid stigma at 12 h after pollination. pt: pollen tube; Ca: callose, bar = 100 μm.

**Figure 2 plants-14-01381-f002:**
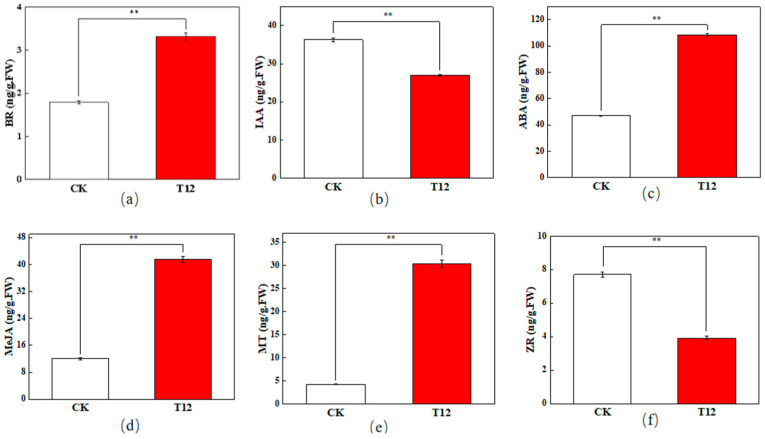
Endogenous hormone contents in stigmas 12 hours after pollination. (**a**) brassinosteroid (BR) content; (**b**) indole-3-acetic acid (IAA) content; (**c**) abscisic acid (ABA) content; (**d**) methyl jasmonate (MeJA) content; (**e**) melatonin (MT) content; (**f**) zeatin riboside (ZR) content. Red and white bars represent endogenous hormone contents in cross-pollinated and self-pollinated stigmas at 12 h after pollination, respectively. **: very significant difference (*p* < 0.01). Error bars represent mean ± standard deviation (SD).

**Figure 3 plants-14-01381-f003:**
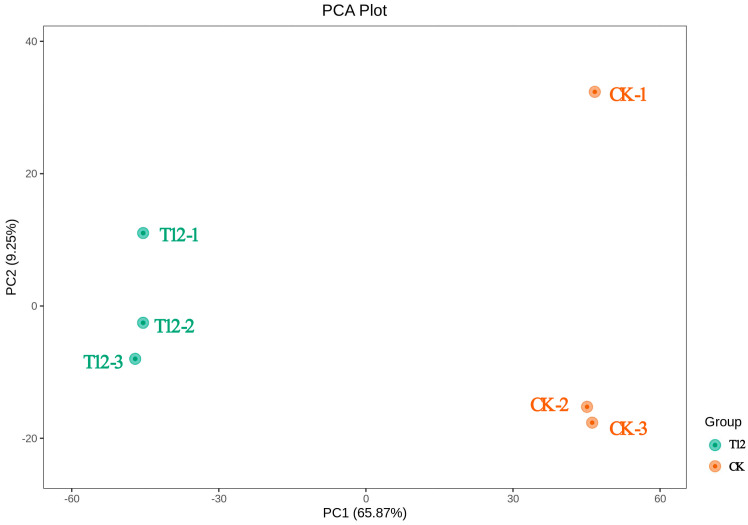
Results of PCA analysis of tree peony stigma metabolites 12 h after pollination. Green dots represent the cross-pollinated group (T12) and yellow dots represent the self-pollinated control group (CK); percentages are the variance explained by each principal component.

**Figure 4 plants-14-01381-f004:**
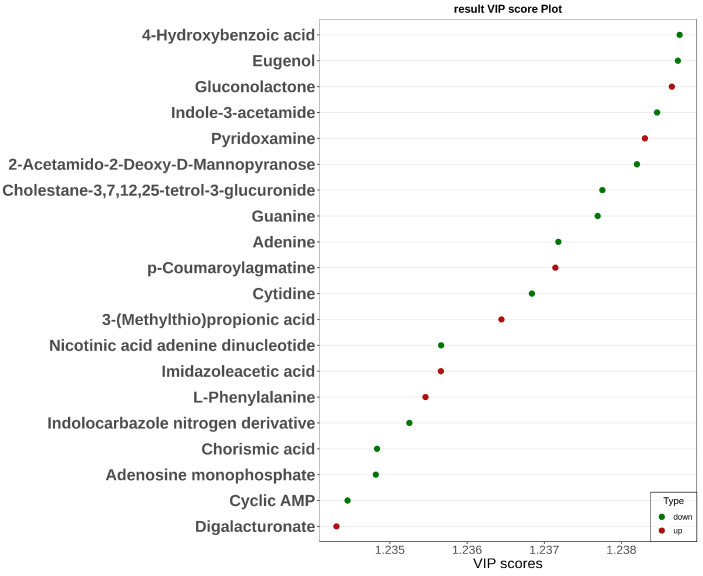
Graph of VIP values of differential metabolites in the KEGG pathway. The horizontal coordinate shows the VIP values; the vertical coordinate shows the differential metabolites; red denotes up-regulated differential metabolites and green denotes down-regulated differential metabolites.

**Figure 5 plants-14-01381-f005:**
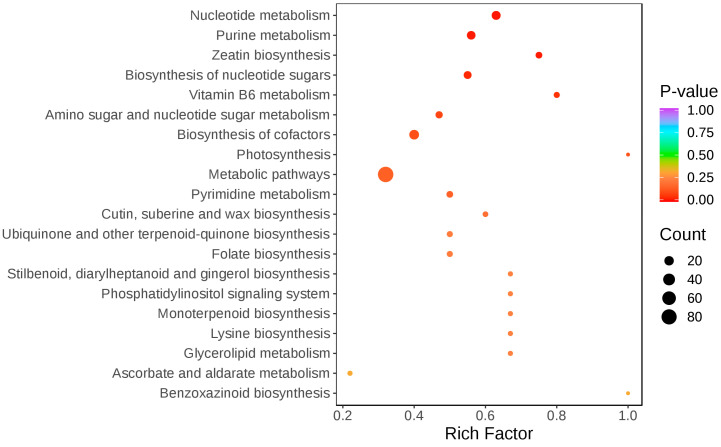
KEGG classification and enrichment plot of differential metabolites. The horizontal coordinate indicates the corresponding Rich Factor of each pathway, and the vertical coordinate is the name of the pathway (sorted according to *p*-value). The color of the dot is the size of the *p*-value, the redder the more significant the enrichment. The size of the dot represents the number of different metabolites enriched.

**Figure 6 plants-14-01381-f006:**
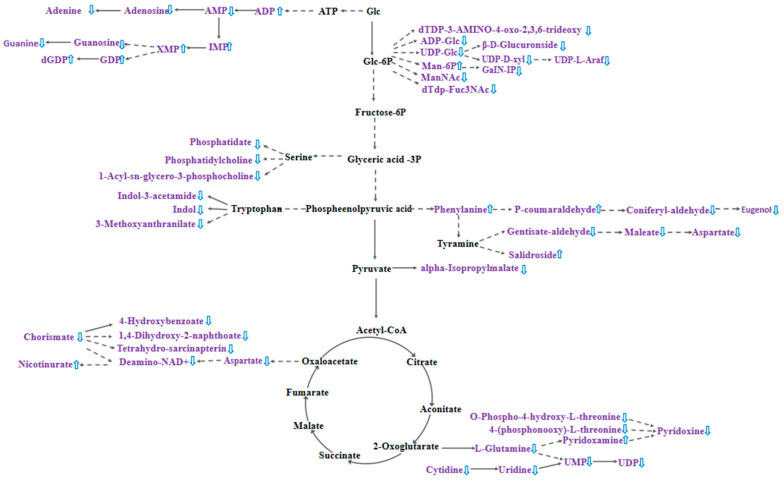
Effect of cross-pollination on stigma metabolic pathways. Metabolites identified in purple text are differential metabolites, up and down direction of blue arrows indicate up-regulation or down-regulation, respectively, of metabolites in T12 compared to CK, and solid and dashed lines connecting the arrows indicate a direct response or an indirect response, respectively.

**Figure 7 plants-14-01381-f007:**
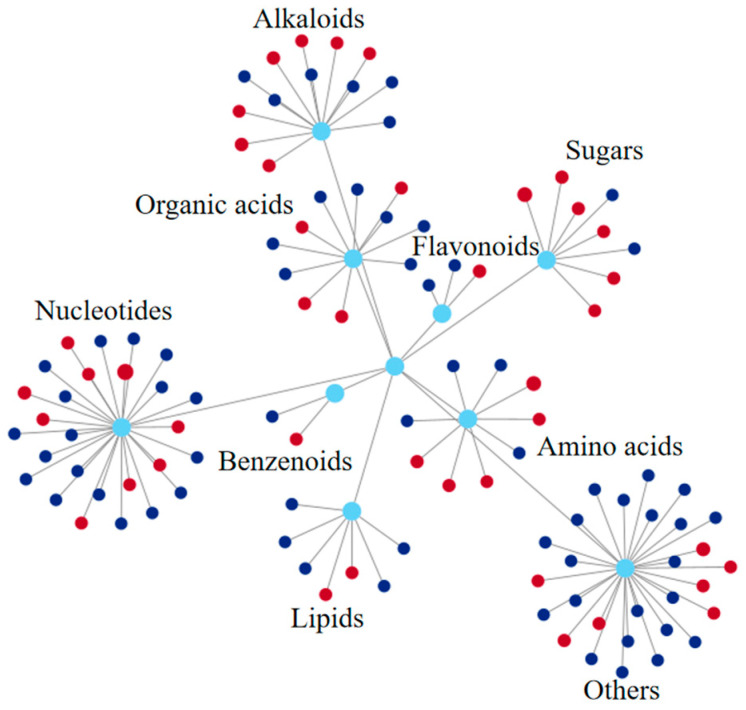
Comprehensive functional/pathway network diagram of differential metabolites in the stigma KEGG pathway. Each node in the figure represents a KEGG pathway-annotated differential metabolite. The diameter of the circles is proportional to the fold change (FC) value. Red color indicates a positive log2FC value, dark blue indicates a negative log2FC value, and light blue indicates metabolite classification.

## Data Availability

Data are contained within the article and Appendix A.

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
