# Peer review of "Non-Target Metabolomics Reveals Changes in Metabolite Profiles in Distant Hybrid Incompatibility Between *Paeonia* sect. *Moutan *and* P. lactiflora"

_plants, 2025, doi:10.3390/plants14091381_

Round 1

Reviewer 1 Report

Comments and Suggestions for Authors

The manuscript describes changes in the metabolism of pollinated stigma of Peonies to identify hybridization barriers in peony breeding. The subject is interesting, however, the data presented must be improved for publication.

Methods:

Metabolite identification: How was the extraction? Solvent? Quantities? There is no mention of MS instrumentation used for the analysis. Brand? Technology?

Data processing is also missing key information. So many acronyms – what is KNN? SVR method? DNA method? How many molecular features were detected in total? Is the differential list representative of this total?

Results:

Figure legends must be self-explanatory – first time mentioning species, then use the whole name, explain acronyms.

Fig. 2. P-value of ANOVA?

Metabolome analysis:

Line 164 – “caused” is not the ideal verb there – “explained” better. PC1 and PC2 will always be the ones explaining most of the variation among samples – it’s how the analysis works.

Fig. 3. I don’t think you need to explain PC1 and PC2 as you did in the legend figure. Key info about the samples is missing.

Fig. 4 – data is presented but not discussed at all. Maybe not relevant then and should be excluded?

Line 197: “Nucleotides are the most abundant metabolites in the stigma” – where does this come from? I cannot see this in figure 7.

You have many figures and not much discussion about what they are showing.

Changes in different classes:

you present the differential data in fold-change (FC) and for the downregulated ones these values are in fractions (as in gene expression datasets). Commonly in metabolomics, these are presented as negative values, instead.  Another point is that I cannot get from your methods how you calculated the fold change. Was the p-value also taken into account?

Subsection titles: “changes in differential sugars” – it seems to me change and differential are redundant. I’d suggest “changes in sugars” or just the class name “sugars”

Line 242: why excess d-mannose and galactose may inhibit pollen germination??

Lines 310 and 311: both lists are for downregulated

Line 327: you don’t mention quercetin is downregulated and then in line 328 you say it is.

For transparency, the complete list of differential metabolites and the data used for their identification should be published as supporting information.

Comments on the Quality of English Language

Please check the whole manuscript for grammar mistakes. e.g. line 194: " number of differentially metabolites." line 219: "metabolism metabolites" 

Author Response

Dear Reviewer:

Thank you for your comments, those comments are all valuable and very helpful for revising and improving our paper, as well as the important guiding significance for our research. According to your comments, we tried our best to improve the manuscript and made some changes in the manuscript. These changes will not influence the content and framework of the paper. And here we did not list the changes but marked in red in revised paper.

We appreciate for reviewers’ warm work earnestly, and hope that the correction that the correction will meet with approval.

Once again, thank you very much for your comments and suggestion. The following part is the point-by-point responses to the reviewer:

The manuscript describes changes in the metabolism of pollinated stigma of Peonies to identify hybridization barriers in peony breeding. The subject is interesting, however, the data presented must be improved for publication.

Methods:

Metabolite identification: How was the extraction? Solvent? Quantities? There is no mention of MS instrumentation used for the analysis. Brand? Technology?

Response: Thank you for your valuable comments. We have carefully revised the manuscript to address your concerns. In the revised version, we have added detailed information about the sample extraction method, as well as information about the solvents used and the quantities. In addition, we have added detailed information about the MS instrumentation, such as the brand and technology used for the analysis. Please refer to the revised version of the paper for details. Thank you.

Data processing is also missing key information. So many acronyms - what is KNN? SVR method? DNA method? How many molecular features were detected in total? Is the differential list representative of this total?

Response: Thank you for your valuable comments. We have carefully revised the manuscript to address all your concerns. Regarding data processing, we have now supplemented the missing key information and added detailed explanations for the acronyms. For "KNN," it refers to K-Nearest Neighbors. For "DNA method," we were specifically referring to the analysis of MtDNA, which is Mitochondrial DNA. As for the "SVR," we have clarified its specific meaning in the revised text. Additionally, we have specified the total number (3853 metabolites) of detected molecular features and explain in detail the screening criteria for the discrepancy list (1242 differential metabolites). All the relevant modifications and supplements have been marked in red in the revised manuscript. Thank you.

Results:

Figure legends must be self-explanatory - first time mentioning species, then use the whole name, explain acronyms.

Response: Thank you for your valuable suggestion. We have carefully revised the figure captions to enhance their self-explanatory nature. In Figure 1, we have replaced the abbreviation "P. ostii ‘Fengdanbai’" with the full name "Paeonia ostii ‘Fengdanbai’" and provided clear explanations for all relevant abbreviations. All the relevant modifications and supplements have been marked in red in the revised manuscript. Thank you.

Fig. 2. P-value of ANOVA?

Response: Thank you for raising this important issue. We have supplemented the p-values from the ANOVA analysis in the revised Figure 2. Please refer to the revised version of the paper for details. Thank you.

Metabolome analysis:

Line 164 - “caused” is not the ideal verb there -“explained” better. PC1 and PC2 will always be the ones explaining most of the variation among samples - it’s how the analysis works.

Response: Thank you for your valuable suggestion. We have replaced "caused" with "explained", Thank you.

Fig. 3. I don’t think you need to explain PC1 and PC2 as you did in the legend figure. Key info about the samples is missing.

Response: Thank you for your valuable suggestion. We have revised the figure note of Figure 3 based on your comments. We have removed the unnecessary explanation of PC1 and PC2 from the figure legend as you suggested. Additionally, we have added the key information about the samples that was previously missing. Thank you.

Fig. 4 - data is presented but not discussed at all. Maybe not relevant then and should be excluded?

Response: Thank you for pointing out the issue with Fig. 4. We sincerely appreciate your careful review. We have changed the content and title of Fig. 4 to "Graph of VIP values of differential metabolites in the KEGG pathway" because our research mainly focuses on metabolites within the KEGG pathway. This modification makes the figure more relevant and aligned with the core of our study. Moreover, we have added a detailed discussion about the data in Fig. 4 in the revised manuscript. Thank you.

Line 197: “Nucleotides are the most abundant metabolites in the stigma” – where does this come from? I cannot see this in figure 7.

Response: Thank you for your valuable comments. In response to your questions, we have revised the text to make it more accurate and consistent with what is presented in figure 7. The previous statement about nucleotides being the most abundant metabolite in the stigma was indeed inaccurate. We have now updated it to read “As shown in figure 7, 108 of the 1242 different metabolites detected in the column head were annotated to the KEGG pathway. Of these, nucleotide metabolites were the most numerous”. In addition, we have added to the legend of figure 7 to make it clear that each node represents a differential metabolite annotated to the KEGG pathway.

You have many figures and not much discussion about what they are showing.

Response: Thank you for your valuable comments. We have carefully revised the discussions related to Figure 4, Figure 5, Figure 6, and Figure 7 in the manuscript. In the revised paper, we provided more detailed interpretations of the data presented in these figures, explaining their significance, implications, and how they contribute to our overall research findings. Thank you.

Changes in different classes:

you present the differential data in fold-change (FC) and for the down-regulated ones these values are in fractions (as in gene expression datasets). Commonly in metabolomics, these are presented as negative values, instead. Another point is that I cannot get from your methods how you calculated the fold change. Was the p-value also taken into account?

Response: Thank you for your interest and valuable comments. we have changed the fold-change (FC) values to log2FC. This conversion allows negative log2FC values to represent down-regulated metabolites, which is consistent with common practice in metabolomics. The revised data have been updated throughout the manuscript.

In addition, we have added a detailed description of the calculation method for FC in the "Methods" section. Additionally, we have incorporated the p-value (p < 0.05) into our analysis to further validate the significance of the results. Thank you.

Subsection titles: “changes in differential sugars” - it seems to me change and differential are redundant. I’d suggest “changes in sugars” or just the class name “sugars”

Response: We sincerely appreciate your valuable suggestion. As recommended, we have revised the subsection title from "changes in differential sugars" to "changes in sugars" for better clarity and conciseness. At the same time, to ensure uniformity throughout the text, we have revised other similar subsection headings accordingly. Thank you.

Line 242: why excess d-mannose and galactose may inhibit pollen germination??

Response: Thank you for your valuable suggestion. Our original formulation is indeed not sufficiently rigorous. We revised it to "In Arabidopsis, excess d-mannose and galactose inhibit pollen germination by interfering with sugar metabolism [40,41]. In the present study, we found that mannose content was significantly up-regulated in peony stigmas, which may produce a similar inhibitory effect and thus adversely affect pollen germination.” Thank you.

Lines 310 and 311: both lists are for down-regulated

Response: Thank you for reminding, we are very sorry for the error. We have promptly revised the content in Lines 310 and 311 to ensure accuracy according to your suggestion. Thank you.

Line 327: you don’t mention quercetin is down-regulated and then in line 328 you say it is.

Response: Thank you for your keen observation. We have added a clarification of this inconsistency. In the revised manuscript, we explicitly mention taxifolin as a down-regulated metabolite. In addition, we have added a reference to demonstrate that taxifolin and quercetin have similar chemical structures. This addition clarifies the relationship between the two compounds and provides a more plausible explanation for the reference to quercetin below. Thank you.

For transparency, the complete list of differential metabolites and the data used for their identification should be published as supporting information.

Response: Thank you for your suggestions. We fully agree with the importance of transparency. We will upload an attachment along with the revised manuscript, which contains the supplementary information requested, including the complete list of differential metabolites and the relevant identification data, to enhance transparency as suggested. Thank you.

Please check the whole manuscript for grammar mistakes. e.g. line 194: " number of differentially metabolites." line 219: "metabolism metabolites"

Response: Thank you for pointing out these issues. We have carefully revised the manuscript. In line 194, "number of differentially metabolites" has been corrected to "number of differential metabolites." As for line 219, "metabolism metabolites" has been amended to " metabolites." We have also conducted a comprehensive grammar check throughout the entire manuscript to ensure its accuracy and quality. Thank you.

Reviewer 2 Report

Comments and Suggestions for Authors

Dear Editor, the ms brings new information about changes in metabolite profiles in distant hybrid incompatibility between Paeonia sect. Moutan and P. lactiflora. The ms was well prepared and authors presented solid conclusions. All my comments are in the attached file. After a minore revision, the ms can be considered for publication in Plants.

Author Response

Dear Reviewer:

Thank you for your comments, those comments are all valuable and very helpful for revising and improving our paper, as well as the important guiding significance for our research. According to your comments, we tried our best to improve the manuscript and made some changes in the manuscript. These changes will not influence the content and framework of the paper. And here we did not list the changes but marked in red in revised paper. Please refer to the uploaded PDF for the point-by-point responses to the reviewer.

We appreciate for reviewers’ warm work earnestly, and hope that the correction that the correction will meet with approval.

Once again, thank you very much for your comments and suggestion.

Response to reviewer ’s comments:

Change  "Azalea" to "azalea"

Response: Thank you for your valuable suggestion. We have corrected "Azalea" to "azalea" on line 54. Thank you.

In our previous study, we found apparent pollen-pistil incompatibility between in the inter-sectional hybrid between P. ostii ‘Fengdanbai’ and P. lactiflora ‘Red Sara’. reference?

Response: Thank you for your valuable suggestion. We have added the relevant reference to support the statement. The reference has been inserted and highlighted in red in the revised manuscript for easy identification. Thank you.

units must be separated from the values (e.g. 12 hours) along the whole text, including Tables and Figures.

Response: Thank you for your careful review and this important suggestion. We have thoroughly revised the entire manuscript, including all tables and figures, to ensure that units are properly separated from the values (e.g., "12 h"). All relevant sections have been carefully checked and corrected. Thank you.

The font size of x and y axis should be enlarged in Figure 2 !

Response: Thank you for your valuable suggestion. We have increased the font size of the x and y axes in Figure 2 as you suggested. The revised figure has been incorporated into the manuscript. Thank you.

Change “down” to “Down”

Response: Thank you for your valuable suggestion. We have capitalized “down” to “Down” in the figure note to match the capitalization of “Up” and ensure consistency. The revised figure note has been updated in the manuscript. Thank you.

Materials: Provide information about the period, including season and year, that the trials were carried out. What was the source of the genetic materials used in all trials?

Response: Thank you for your valuable suggestion. We have supplemented the manuscript with information on the time period including season and year of the trials. Thank you for your constructive comments, which have enhanced the comprehensiveness of our research description. Thank you.

What was the source of the genetic materials used in all trials?

Response: Thank you for your valuable suggestion. We have clearly indicated the source of the genetic materials used in all trials. Thank you for your constructive comments, which have enhanced the comprehensiveness of our research description. Thank you.

Change “ultra -low“ to “ultra-low”.

Response: Thank you for your valuable suggestion. We have corrected the hyphenation of "ultra -low" to "ultra-low" as suggested. Thank you for pointing out this error. Thank you.

use either h or hours along the whole manuscript.

Response: Thank you for your valuable suggestion. We have unified the usage of time units throughout the manuscript, consistently using either "h" as per the suggestion. Thank you for your careful review.

2.2.2. Determination of hormone content

Change “hormone“ to “endogenous hormone”

Response: Thank you for your valuable suggestion. We have changed "hormone" to "endogenous hormone" in the section of "Determination of hormone content" as you suggested. Thank you.

The differential metabolites detected were subjected to data pre-processing and data quality control, and the obtained high-quality data were statistically analyzed to generate clustered heat maps, KEGG pathway enrichment classifications and scatter plots. This line should be a separated paragraph.

Response: Thank you for your valuable suggestion. We have separated the sentence "The different metabolites detected were subjected to data pre-processing and data quality control, and the obtained high-quality data were statistically analyzed to generate clustered heat maps, KEGG pathway enrichment classifications and scatter plots." into a separate paragraph as you suggested. Thank you.

Round 2

Reviewer 1 Report

Comments and Suggestions for Authors

no comments